# Comparison of Commercial Poultry Semen Extenders Modified for Cryopreservation Procedure in the Genetic Resource Program of Czech Golden Spotted Hen

**DOI:** 10.3390/ani12202886

**Published:** 2022-10-21

**Authors:** Kristýna Petričáková, Martina Janošíková, Martin Ptáček, Lukáš Zita, Filipp Georgijevič Savvulidi, Agnieszka Partyka

**Affiliations:** 1Department of Animal Sciences, Faculty of Agrobiology, Food and Natural Resources, Czech University of Life Sciences Prague, Kamýcká 129, 165 00 Praha, Czech Republic; 2Faculty of Veterinary Medicine, Department of Reproduction and Clinic of Farm Animals, Wroclaw University of Environmental and Life Sciences, pl. Grunwaldzki 49, 50-366 Wroclaw, Poland

**Keywords:** cryoprotectant, flow cytometry, gene reserve, rooster, sperm evaluation, spermatozoa

## Abstract

**Simple Summary:**

The Czech Golden Spotted Hen is the only original Czech hen breed included in the National Genetic Reserve Program. The objective of this program is to preserve the genetic resources of native breeds in the Czech Republic through reproductive biotechnologies, such as artificial insemination or the cryopreservation of sperm. In this study, three commercial liquid storage extenders supplemented with a 9% N-methylacetamide cryoprotectant were investigated. Using a defined methodology, modified Poultry media^®^ and Raptac^®^ extenders were identified as promising tools for the cryopreservation procedure. Thus, an important strategy for the National Genetic Reserve Program of the Czech Golden Spotted Hen was suggested.

**Abstract:**

Spermatozoa cryoconservation represents an important strategy for partial in vitro or rescue programs designed for threatened livestock populations. The procedure for the semen cryopreservation of the Czech Golden Spotted Hen was proposed due to the lower fertilization rate of poultry semen compared to mammalian species. The aim of this study was to compare commercial extenders designed for liquid storage preservation with the use of a predefined cryoprotectant, and, thus, to propose an important tool for the procedure of the semen cryopreservation of the Czech Golden Spotted Hen. Ejaculates were sampled from four roosters during five semen collection days. The samples were frozen in Poultry media^®^, Raptac^®^ and NeXcell^®^ extenders supplemented with a 9% N-methylacetamide (NMA) cryoprotectant. Sperm parameters of the total motility (MOT; %), plasma membrane and acrosome intactness (PAI; %), plasma membrane damage (%), acrosome damage (%) and cells with plasma membrane and acrosome damage (%) were assessed using a mobile mCASA analyzer and flow cytometer after the cryopreservation of the insemination doses (IDs). For Poultry media^®^ (PAI = 51.11%; MOT = 23.58%) and Raptac^®^ (PAI = 52.04%; MOT = 23.13%) extenders with the addition of an NMA cryoprotectant, the comparable results were detected after thawing. For NexCell^®^ media, the results were poor (PAI = 7.07%; MOT = 3.83%). Our results indicated two extenders suitable for the cryopreservation procedure, with the applied modification.

## 1. Introduction

The Czech Golden Spotted Hen (CGS), the only original Czech hen breed classified among genetic resources, is the last classical representative of original Central European peasant hens of the light type. It is typically chosen for its specific coloring and suitability for free-range or organic farming conditions, as Kraus et al. [1,2] described. Additionally, the CGS has been included in the Native Breeds Protection Program and protected under the national rescue program since 1996 [3]. The important part of this program represents an in vitro reservoir of long-term cryopreserved insemination doses (IDs).

The success of a cryopreservation procedure is determined with a suitable extender. Extenders can be defined as buffered salt solutions used to prolong the viability of good-quality semen. The main advantage of commercial extenders is their availability, and standardized composition and application. Unfortunately, cryopreservation is not performed in poultry and no media are standardized for this purpose. For this reason, supplementation with cryoprotectants that preserve the cells during the freezing process represents an important strategy in the modification process [4,5]. Some of the most used cryoprotectants include glycerol [6,7], N-methylacetamide (NMA) [8], dimethylacetamide [9,10,11,12] or dimethyl sulfoxide [13,14]. The glycerol addition to IDs reduces the fertilizing ability of sperm. Glycerol concentrations in the IDs as high as 2% result in the complete infertility of poultry spermatozoa. For that reason, it is necessary to remove it by washing it off before insemination [15,16]. This procedure, however, is detrimental due to the osmotic pressure, which causes mechanical damage to the poultry sperm and, thus, reduces the fertilizing capacity of the IDs. The investigation of other cryoprotectants, without such a toxic effect on spermatozoa, is an urgent task for current research. Recently, NMA was successfully studied for the cryopreservation of rooster semen [5]. This approach was supported by results published by Santiago-Moreno et al. [17] and Mosca et al. [18], where a higher sperm survival rate was detected for IDs conserved in an extender with an NMA addition compared to dimethylacetamide. The NMA supplementation in the semen extender should be in the range of 6% [19] to 9% [8,20,21,22]. Additionally, the optimal NMA concentration at a 9% level was also identified by Petricakova et al. (unpublished data) for the cryopreservation procedure in the CGS breed.

The aim of the study was to compare commercial extenders supplemented with the defined concentration of cryoprotectant, and to suggest an important tool for the procedure of the semen cryopreservation of the native Czech poultry breed.

## 2. Materials and Methods

### 2.1. Animals and Housing

All procedures performed with animals were in accordance with the Ethics Committee of the Central Commission for Animal Welfare at the Ministry of Agriculture of the Czech Republic (Prague, Czech Republic), and were carried out in accordance with Directive 2010/63/EU for animal experiments. All investigations and handlings were conducted using normal breeder practices in accordance with official laws in the Czech Republic (Animal Protection Against Cruelty Act; Act No.246/1992 Sb.).

Four roosters of the Czech Golden Spotted Hen, all representatives of the important bloodline with a demonstrable Certificate of Fowl Origin were included in this study. The average age of the roosters was 68 weeks. The males were kept in individual cages under controlled environmental conditions (air temperature = approx. 20 °C; air humidity = 50–60%) throughout the whole study in the Demonstration and Experimental Centre of the Czech University of Agriculture in Prague (50°07′47.6″ N 14°22′07.0″ E). The day/night photoperiod was 14/10 h during the experiment. The roosters were fed a complete compound mixture produced for the Czech University of Life Sciences in Prague (Sehnoutek a synové s.r.o.; 15.00% crude protein, 11.56 MJ/kg of metabolizable energy). Feed and water were available to the animals ad libitum throughout the experiment.

### 2.2. Semen Collection and Insemination Dose Processing

Rooster semen was collected twice a week during January. There was an interval of 2 to 3 days between the subsequential collection day. Semen collection was performed using the dorsoabdominal massage technique explained in [23]. The collection was always performed by the same person at 8 AM. The collected ejaculate was stored at 5 °C until processing.

The ejaculate volume and color were macroscopically evaluated immediately after collection and recorded directly from the semen collection tube. The volume of semen was expressed as a microliter (μL). All semen samples that passed initial requirements (the minimum value of spermatozoa concentration was set at 2 × 10^9^ cells/mL) were mixed and diluted in commercially available extenders, including Poultry media^®^ without antibiotics (IMV Technologies, L’Aigle, France), Raptac^®^ (AMP-Lab, GmbH, Germany) and NeXcell^®^ (IMV Technologies, L’Aigle, France), to a final concentration of 100 × 10^6^/mL cells. All samples were supplemented with an N-methylacetamide (NMA) cryoprotectant at a defined concentration of 9% [8,11]. Control IDs without NMA were also performed, but due to the absence of a cryoprotectant, complete cryodestruction was achieved, so these results were not included in this study.

After reaching equilibration at 5 °C for 1 min [18], diluted semen samples were transferred into 0.25 mL French straws (IMV Technologies, L’Aigle, France) and sealed with sealing powder (IMV Technologies, L’Aigle, France). The IDs were frozen using liquid nitrogen vapor (the straws were placed 5 cm above the surface) for 10 min [24]. The freezing of the IDs was performed in a polystyrene mobile freezing box (commercial freezing box Minitube, GmbH). Subsequently, the straws were directly immersed in liquid nitrogen for long-term preservation. Finally, the straws were transferred to a nitrogen tank (−196 °C) and kept at this temperature for three months before use.

### 2.3. Evaluation of the Quality of Insemination Doses

The thawing procedure was performed in a water bath tempered at 5 °C for 100 s [8] to assess the quality of the IDs. Three IDs from each variation (modified extenders within the semen collection day) were thawed as described and pooled. Afterward, the exact aliquots of pooled sperm were assessed in three technical replicates, using a computer-assisted sperm analysis and flow cytometry, as described below. For the subsequent statistical evaluation, all the data on technical replicates were averaged. The key functional spermatozoa parameter of the total motility (MOT) was assessed with a mobile mCASA analyzer (iSperm^®^, Aidmics Biotechnology, Taipei City, Taiwan) after thawing using the iSperm Poultry 5 (iSperm^®^, Aidmics Biotechnology, Taipei City, Taiwan) application. IDs were diluted with the same extender that was used to dilute the semen before freezing to the mCASA manufacturer’s recommended concentration of 30 × 10^6^ sperm/mL. The diluted sample was dropped in a volume of 7 μL onto a special analysis disposable chamber that was fixed on the lens; then, the total motility was evaluated using the software.

The percentages of spermatozoa with intact plasma membranes and acrosomes (PAI, %), with damaged plasma membranes (PMD, %) and with damaged acrosomes (ACRD, %) were determined using flow cytometry assay. Before flow cytometry, IDs were thawed as described above. Thawed IDs were diluted with phosphate-buffered saline (Sigma Aldrich, St. Louis, MO, USA) to a final concentration of 10 × 10^6^/mL, and stained with a mix of fluorescent dyes at 38.5 °C for 15 min (in the dark). The following dyes were used in the mix (final concentrations given): 16.6 µg/mL Hoechst-33342 (H-342) to identify the presence of DNA; 13.3 µg/mL propidium iodide (PI) to detect plasma membrane damage; 0.83 µg/mL PNA lectin from Arachis hypogea (PNA-FITC) to assess acrosome damage. H-342 and PI were purchased from Sigma Aldrich (St. Louis, MO, USA) and PNA-FITC from Thermo Fisher Scientific (Waltham, MA, USA). Flow cytometric parameters were assessed with NovoCyte 3000^®^ flow cytometer (Acea Biosciences, Agilent, Santa Clara, CA, USA). The flow cytometer was equipped with violet (405 nm), blue (488 nm) and red (640 nm) lasers and appropriate optical filters for the detection of emitted fluorescence signals. H-342 could be successfully excited with the violet (405 nm) laser (Martinez-Pastor et al. 2010). NovoExpress software, v1.3.0 (Acea Biosciences, Agilent, Santa Clara, CA, USA) was used for data acquisition. The same software was used to analyze the acquired flow cytometry data. No compensation was required with the optical filter setup used. The gating strategy was presented in Figure 1.

### 2.4. Statistical Evaluation

All statistical evaluations were performed in the SAS statistical program (SAS/STAT v9.4. (SAS Institute Inc., Cary, NC, USA). The general linear model (GLM) procedure was used to analyze variables after a check of residual distribution (residual QQ plot) [14,25]. A statistical model with two fixed effects of the day of semen collection (5 classes with 18 observations per class) and modified extenders (3 classes with 30 observations per class) was applied to cover the variability among dependent variables. The following model equations were fitted for this estimation:Y_ijk_= μ + DAY_i_ + EXT_j_ + e_ijk_
where: Y_ijk_—dependent variable (total motile sperm after thawing, sperm with plasma membrane and acrosome intactness after thawing, sperm with plasma membrane damage after thawing, sperm with acrosomal damage after thawing, sperm with plasma membrane and acrosomal damage after thawing); μ—mean value of dependent variable; DAY_i_—fixed effect of the semen collection day (i = 10 January, *n* = 18; i = 14 January, *n* = 18; i = 17 January, *n* = 18; i = 20 January, *n* = 18; i = 24th January, *n* = 18); EXT_j_—fixed effect of the modified extender (j = Poultry media^®^ with 9% of NMA supplementation, *n* = 30; j = Raptac^®^ with 9% of NMA supplementation, *n* = 30; j = NeXcell^®^ with 9% of NMA supplementation *n* = 30); e_ijk_—residual error.

The significance of the differences between the semen collection days and the individual modified extenders was tested using the Tukey–Kramer test. A significance level *p* < 0.05 was used to evaluate the differences between groups.

## 3. Results

The summary statistics of the database structure are reported in Table 1. The average value of the motility, plasma membrane, and acrosome intactness in thawed IDs, regardless of the impact of the fixed effect of the day and the fixed effect of the added modified extender, was 17.23% and 36.08%, respectively.

The model equations were statistically significant (*p* < 0.001) and explained 60.11% to 97.43% of the variability of the monitored parameters. Additionally, both factors, the day of semen collection and the extender, were significant for all monitored parameters (*p* < 0.05).

Results related to days of semen collection are reported in Table 2. In general, a stronger statistical conclusiveness across days of semen collection was detected for all flow cytometric parameters to the MOT. The maximal PAI difference reached 13.80% (first vs. fourth day), while the PMD or ACRD maximal differences were 18.40% or 9.90%, respectively.

In general, comparable MOT results were detected for IDs frozen in the Poultry media^®^ (23.58%) and Raptac^®^ (23.13%) modified extenders supplemented with 9% of NMA. On the other hand, the motility of the IDs frozen in a modified NeXcell^®^ extender after thawing reached significantly lower values—compared to both other extenders modified in the same manner (Figure 2).

Comparable results for spermatozoa PAI were demonstrated between the modified Poultry media^®^ (51.11%) and Raptac^®^ (52.04%) extenders analyzed using flow cytometry as well (Table 3). IDs frozen in NeXcell^®^ with the cryoprotective agent reached a significantly lower PAI (7.07%) compared to both investigated modified extenders. This was supported by the significantly highest PMD (87.03%) of spermatozoa frozen in NeXcell^®^ with the 9% NMA supplement. Contrary, no significant differences were detected for the percentages of spermatozoa with damaged acrosomes among all evaluated modified extenders. Numerically higher acrosomal damage was, however, observed in Poultry media^®^ (3.36%) followed by Raptac^®^ (2.87%) and NeXcell^®^ (2.59%), all supplemented with the defined cryoprotective agent.

## 4. Discussion

The freezing of semen is an important tool for preserving the genetic resources of threatened animals for the possibility of preserving large numbers of IDs [26]. This procedure is very important in breeding schemes in different livestock for spreading gene flow across the population and to use genetically valuable animals. However, the fertilization rate of poultry inseminated with frozen semen is dramatically lower compared to fresh semen [27]. According to our knowledge, there are no standardized extenders for poultry semen cryopreservation. An important task of current research is to investigate extenders modified for the cryopreservation of rooster spermatozoa. The aim of this study was to analyze frozen–thawed spermatozoa preserved in commercial liquid storage extenders supplemented with 9% NMA concentration.

The importance of factors such as the semen collection day on CASA and flow cytometric parameters of rooster sperm after thawing was previously reported on by Long et al. [28], Sonseeda et al. [29], Long et al. [30], Meamar et al. [31] and Rakha et al. [32], even though the roosters were kept in the same conditions throughout the experiments. Significant differences among semen collection days were detected for all the evaluated CASA and flow cytometric parameters in our study as well. These differences were obvious even in an approx. 3-week time interval, and among the same roosters managed under the same conditions. This demonstrated that the selection of a particular semen collection day is a very important factor and has to be considered when processing semen IDs.

During semen cryopreservation, the type of cryoprotectant used also plays a very important role. Recently, NMA was successfully studied for the cryopreservation of rooster semen [8,17,18]. This was supported by results published by Lee et al. [17] and Mosca et al. [18], when a higher sperm survival rate was detected for IDs conserved in an extender with an NMA addition compared to dimethylacetamide.

Unfortunately, there are only a few studies investigating the effect of NMA cryoprotectants using in vitro assays in the literature. Mosca et al. [19] compared 6% and 9% concentrations of NMA in the Lake semen extender. Their results indicated that the decrease in NMA concentration from 9 to 6% improved sperm quality after freezing/thawing in Hy-Line White chicken. In comparison with our study, Mosca et al. [19] obtained, in general, lower results of PAI for both 6% NMA (50.76%) and 9% NMA (36.6%) supplements. On the other hand, they found a higher sperm motility by using 6% NMA (52.3%) than 9% NMA (35.5%), compared our results. Contradictorily, Pérez-Marín et al. [33] obtained a very impressive sperm motility of 65% using 9% NMA in the Combatiente Español breed. Their results were, thus, higher in comparison to our results in all the other available studies as well. We used a 9% NMA concentration based on the pilot study, indicating its suitability for freezing the CZH breed (Petricakova et al., unpublished data). Differences between our results to other authors might be related to the different breeds or differences in ID handling and processing. The generalization of other breeds for optimizing ID cryopreservation approaches is a perspective for further research as well.

A very important step to improve the cryopreservation process of rooster semen is the artificial insemination of hens, as an “absolute” test of any cryopreservation procedure. Sasaki et al. [8] achieved an 89.5% hatchability using 9% NMA (using Yakido roosters and White Leghorn hens). On the other hand, Pérez-Marín et al. [33] found low values of fertility (9.4%) and hatchability (7%) for IDs containing 9% NMA, despite the very perspective motility of the spermatozoa. This indicated that the insemination technique is very informative in terms of feedback on all optimizing cryopreservation procedures. For that reason, the interrelation of in vitro and in vivo assays should be compared and identified as well. This represented a very prospective tool for consecutive research activities.

All the extenders that we used in our study were designed for liquid storage preservation. These extenders naturally contain different supportive substances, such as antioxidants, sugars or vitamins. Unfortunately, the authors were unable to determine the differences between the individual extenders because the exact compositions and contents were unknown. It is potentially possible that some components of the NeXcell^®^ extender interacted negatively with the NMA cryoprotectant. The use of a different cryoprotectant might have resulted in better sperm functional parameters.

In this trial, the best cryoprotective effect was detected for the Poultry media^®^ and Raptac^®^ extenders with 9% of NMA supplementation. On the other hand, NeXcell^®^ with the same modification was less suitable for this purpose. More importantly, the authors were fully aware of the fact that the extenders were modified and tested for cryopreservation, for which they were not primarily intended. We in no way questioned the effect of these extenders for short-term preservation.

## 5. Conclusions

The cryopreservation of rooster spermatozoa is an important strategy for rescue programs of threatened populations of animals (original Czech breed) as for the intensive spread of gene flows across populations. Our results suggested two extenders (Poultry media^®^ and Raptac^®^) would be suitable for the cryopreservation procedure, with an applied modification. These extenders were tested for high qualitative spermatozoa parameters (assessed with an mCASA device and flow cytometry), and they did not differ significantly from each other. Our results suggested a perspective strategy for the cryopreservation procedure of avian spermatozoa.

## Figures and Tables

**Figure 1 animals-12-02886-f001:**
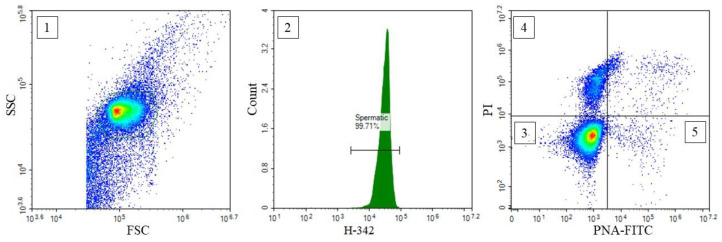
Flow cytometric gating strategy. A cluster of events was initially identified using a side scatter (SSC) versus forward scatter (FSC) bivariate histogram plot (**1**). Spermatic events were identified based on the gating set with the Hoechst-33342 stain (H-342; DNA content) (**2**). Finally, the percentage of spermatozoa with intact plasma membranes and intact acrosome—PAI (**3**)—with damaged plasma membranes—PMD (**4**)—and with damaged acrosomes—ACRD (**5**)—were identified based on the propidium iodide (PI) and Arachis hypogea lectin PNA (PNA-FITC) signal intensities. Illustrative dot plots and histograms are shown.

**Figure 2 animals-12-02886-f002:**
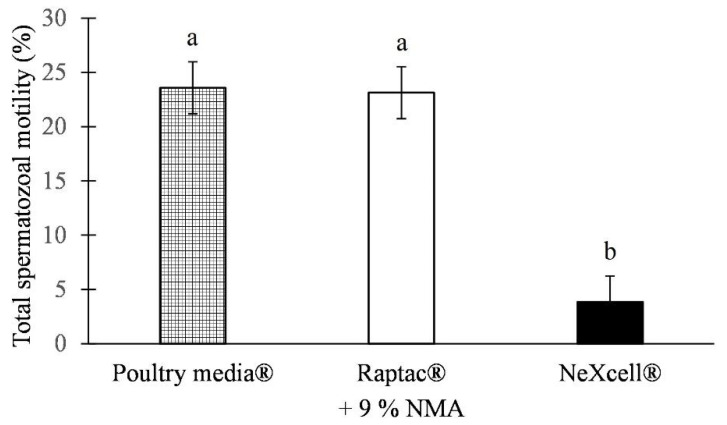
The percentage of the motile post-thaw spermatozoa (MOT) cryopreserved in different commercial extenders supplemented with 9% N-methylacetamide (least square means ± standard error). ^a,b^ Different letters indicate differences between groups within a column (*p* < 0.05).

**Table 1 animals-12-02886-t001:** Summary statistics of database structure for spermatozoa cells diluted in three commercial extenders supplemented with different extenders and 9% NMA after thawing.

Variable	*n*	AM	SD	Minimum	Maximum
MOT	90	17.23	18.52	0.00	68.00
PAI	90	36.08	21.83	3.04	58.65
PMD	90	58.96	21.98	27.50	94.96
ACRD	90	2.95	4.06	0.12	13.56

AM—arithmetic mean; SD—standard deviation; MOT—percentage of total motile sperm after thawing; PAI—percentage of spermatozoa that were not found to have plasma membrane or acrosome disruptions; PMD—percentage of sperm with plasma membrane damage after thawing; ACRD—percentage of sperm with acrosome damage after thawing.

**Table 2 animals-12-02886-t002:** The impact of daily variation on the motility (MOT), plasma membrane and acrosome intactness (PAI), plasma membrane damage (PMD), and acrosome damage (ACRD) of thawed rooster sperm (least square means ± standard error).

DAY	MOT (%)	PAI (%)	PMD (%)	ACRD (%)
10 January (*n* = 18)	22.18 ± 2.89 ^ab^	42.79 ± 0.86 ^a^	44.98 ± 0.98 ^c^	2.73 ± 0.22 ^b^
14 January (*n* = 18)	23.61 ± 2.89 ^a^	38.85 ± 0.86 ^ab^	58.80 ± 0.98 ^b^	0.99 ± 0.22 ^c^
17 January (*n* = 18)	8.39 ± 2.89 ^b^	35.80 ± 0.86 ^b^	63.33 ± 0.98 ^a^	0.25 ± 0.22 ^c^
20 January (*n* = 18)	15.67 ± 2.89 ^ab^	28.98 ± 0.86 ^c^	59.23 ± 0.98 ^b^	10.19 ± 0.22 ^a^
24 January (*n* = 18)	14.39 ± 2.89 ^ab^	37.30 ± 0.86 ^b^	61.42 ± 0.98 ^ab^	0.56 ± 0.22 ^c^

MOT—percentage of total motile sperm after thawing; PAI—percentage of spermatozoa that were not found to have plasma membrane or acrosome disruptions; PMD—percentage of sperm with plasma membrane damage after thawing; ACRD—percentage of sperm with acrosome damage after thawing. ^a–c^ Different letters indicate differences between groups within a column (*p* < 0.05).

**Table 3 animals-12-02886-t003:** Impact of semen extender supplemented with defined cryoprotective agent on the plasma membrane and acrosome intactness (PAI), plasma membrane damage (PMD) and acrosome damage (ACRD) of thawed rooster sperm (least square means ± standard error).

Extender	PAI (%)	PMD (%)	ACRD (%)
Poultry media^®^ + 9% NMA (*n* = 30)	51.11 ± 0.72 ^a^	43.06 ± 0.82 ^b^	3.36 ± 0.18
Raptac^®^ + 9% NMA (*n* = 30)	52.04 ± 0.72 ^a^	42.57 ± 0.82 ^b^	2.87 ± 0.18
NeXcell^®^ + 9 % NMA (*n* = 30)	7.07 ± 0.72 ^b^	87.03 ± 0.82 ^a^	2.59 ± 0.18

NMA—N-methylacetamide supplement; PAI—percentage of spermatozoa that were not found to have plasma membrane or acrosome disruptions; PMD—percentage of sperm with plasma membrane damage after thawing; ACRD—percentage of sperm with acrosome damage after thawing. ^a,b^ Different letters indicate differences between groups within a column (*p* < 0.05).

## Data Availability

The data presented in this study are available on request from the corresponding author.

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
