# Peer review of "Comparison of Commercial Poultry Semen Extenders Modified for Cryopreservation Procedure in the Genetic Resource Program of Czech Golden Spotted Hen"

_animals, 2022, doi:10.3390/ani12202886_

Round 1
Reviewer 1 Report
This paper is a valuable study to improve the freezing extender by supplementation of NMA, instead of toxic glycerol, to cryopreserve Czech Golden Spotted Hen spermatozoa. The assays using CASA and Flowcytometry drew reliable results in this paper. Therefore, this article will be useful for avian sperm freezing.
Author Response
Dear Reviewer 1,
thank you for your review on our manuscript. Please find attached file with our responses in enclosure.

Reviewer 2 Report
The authors compare the cryoprotective effect of three commercial extenders supplemented with N-methylacetamide in poultry semen from the Czech golden chicken genetic resource programme. The results show that only 2 out of 3 media used were effective in maintaining viability and integrity of plasma and acrosomal membranes.
Abstract
- I recommend that authors write the keywords in alphabetical order.
- The summary needs a clear initial statement (prior to the general objective) of the problem addressed.
- In addition, the authors should describe the main conclusion from the study
Introduction
- the introduction is well structured and highlights the importance of the study in the area of knowledge.
Materials and methods
- The authors describe the procedures in detail in section “2.2. Semen collection and insemination doses processing”, but they do not detail how long the straws were stored for.
- Did the authors freeze semen straws with the different diluents but without NMA as a control? This control is important to rule out that the combination of NMA and Nexcell medium or extender is the cause of a negative effect on the cells and not the medium itself, or to demonstrate that the addition of NMA has a synergistic positive effect on Poultry media and Raptac, which would justify its use as a complement to the extenders.
- IMV Technologies manufactures different formulations of poultry media, Could the authors give more details on which formula they used?
- The authors state in section 2.3 “IDs were diluted with the appropriate extender to the manufacturer's recommended”. I recommend that the authors describe the diluent used to dilute the samples post thawing. The material and methods section should be described with sufficient detail to allow others to replicate and build on published results.
- which English words do the acronym PAI come from?
- In section 2.4 "Statistical evaluation" the authors describe an equation model for statistical analysis of the results. Can the authors indicate the source of this equation and cite its validation?
- How did the authors assess the distribution of the data?
- Why did the authors not use an anova and post-test in their analysis?
- how many samples of each cock were frozen-thawed and analysed?
- Were the experiments conducted in duplicate or triplicate?
Results
- Table 1 describes several results, but it is not clear whether these are spermatozoa thawed without supplementation (control) or from the different diluents supplemented with NMA. If they are cryopreserved cells in unsupplemented media, this group is not described in the material and methods section.
- The description of the results is confusing, and the description and legends of the tables are inaccurate. I recommend the authors to edit them.
Discussion
- Methodologically the first paragraph of the discussion is a brief description of the rationale for the study ending with the general objective, but in this case the general objective appears in the third paragraph. I recommend the authors to review the order of the paragraphs and edit this section of the manuscript.
- lines 240-255: Although the variable in this study was the different diluents supplemented with the same concentration of NMA, the authors discuss their results with other studies comparing two different concentrations of NMA. This discussion would be more appropriate if unsupplemented controls had been included in this study (not indicated in the material and methods section).
- It would be interesting if the authors could discuss and/or explain the negative results found with the Nexcell diluent or at least hypothesise about it.
Conclusions
- Conclusions are statements that based on the results and discussion allow us to establish something as true, valid, or possible. Therefore, I recommend that authors do not include the objective of the study in this section and ideally improve the wording.
Author Response
Dear Reviewer 2,
thank you for your review on our manuscript. Please find attached file with our responses in enclosure.

Reviewer 3 Report
Thank you very much for giving me an opportunity to review "Comparison of commercial poultry semen extenders modified 2 for cryopreservation procedure in genetic resource program 3 of Czech golden spotted hen". The study is interesting and provides some baseline information of different cryopreservants for preservation of native chicken sperms. The following minor revision should be submitted before publication of this research work.
a. does the number of roosters are not too small to conduct a full length research study on the topic?
b. All abbreviated in the text and the tables must be defined well for easy appraoch.
c. Where there is no significant difference, alphabets must be remvoed.
d. Conclusion is too much lengthy and ambiguous. It should be summerized keeping the main results of your study. No suggestions and recommendations are needed to be given in this part. The best extender must be pointed out.
D. in discussion part, you should mention the reasons due to which there is a difference in the quality of semen due to the presence of different extenders.
Author Response
Dear Reviewer 3,
thank you for your review on our manuscript. Please find attached file with our responses in enclosure.

Round 2
Reviewer 2 Report
The authors responded satisfactorily to all the questions raised and made the suggested modifications. This improved the manuscript considerably.